# Pharmacometabolomics Applied to Personalized Medicine in Urological Cancers

**DOI:** 10.3390/ph15030295

**Published:** 2022-02-28

**Authors:** Filipa Amaro, Márcia Carvalho, Maria de Lourdes Bastos, Paula Guedes de Pinho, Joana Pinto

**Affiliations:** 1Associate Laboratory i4HB—Institute for Health and Bioeconomy, Department of Biological Sciences, Laboratory of Toxicology, Faculty of Pharmacy, University of Porto, 4050-313 Porto, Portugal; mcarv@ufp.edu.pt (M.C.); mlbastos@ff.up.pt (M.d.L.B.); pguedes@ff.up.pt (P.G.d.P.); 2UCIBIO/REQUIMTE, Department of Biological Sciences, Laboratory of Toxicology, Faculty of Pharmacy, University of Porto, 4050-313 Porto, Portugal; 3FP-I3ID, FP-ENAS, CEBIMED, University Fernando Pessoa, 4200-150 Porto, Portugal; 4Faculty of Health Sciences, University Fernando Pessoa, 4200-150 Porto, Portugal

**Keywords:** prostate cancer, bladder cancer, renal cell carcinoma, pharmacometabolomics, biomarkers, treatment response

## Abstract

Prostate cancer (PCa), bladder cancer (BCa), and renal cell carcinoma (RCC) are the most common urological cancers, and their incidence has been rising over time. Surgery is the standard treatment for these cancers, but this procedure is only effective when the disease is localized. For metastatic disease, PCa is typically treated with androgen deprivation therapy, while BCa is treated with chemotherapy, and RCC is managed primarily with targeted therapies. However, response rates to these therapeutic options remain unsatisfactory due to the development of resistance and treatment-related toxicity. Thus, the discovery of biomarkers with prognostic and predictive value is needed to stratify patients into different risk groups, minimizing overtreatment and the risk of drug resistance development. Pharmacometabolomics, a branch of metabolomics, is an attractive tool to predict drug response in an individual based on its own metabolic signature, which can be collected before, during, and after drug exposure. Hence, this review focuses on the application of pharmacometabolomic approaches to identify the metabolic responses to hormone therapy, targeted therapy, immunotherapy, and chemotherapy for the most prevalent urological cancers.

## 1. Introduction

The incidence and mortality of cancer tend to increase over the next years as a reflection of population aging and growth [1,2,3]. Incidence rates broadly vary between sexes and world regions due to differences in lifestyle habits (e.g., diet, nutrition, physical activity), exposure to risk factors, and disparities in quality of cancer prevention, diagnosis, and treatment [4]. The most recent data from GLOBOCAN [2] estimated 19.3 million new cancer cases and 10.0 million deaths worldwide in 2020. Considering the urinary system, prostate cancer (PCa), bladder cancer (BCa), and kidney cancer (KCa) are the most common malignancies. PCa represented the most incident urological cancer in 2020, accounting for more than half of all diagnosed cases. Furthermore, PCa ranked 3rd in the list of most prevalent cancers according to GLOBOCAN [2]. BCa was the 2nd most prevalent urological cancer, ranking in the 11th position among the worldwide most common cancers, while KCa was the 3rd most prevalent urological cancer, ranking in the 15th position among the most common cancers [2]. Renal cell carcinoma (RCC) represented more than 90% of all KCa diagnoses [5] and can be divided according to different histological subtypes into clear-cell RCC (ccRCC), which constitutes 75% of all RCC cases [6], and the remaining 20% included the papillary and chromophobe RCC subtypes, among other rare types [6,7]. Testicular, penile, and urethral cancers are also classified as urological cancers; however, due to their low prevalence worldwide (~1%), they were not included in this review.

Significant progress has been made in the understanding of hallmarks underlying the development of urological cancers and the identification of novel molecular markers to improve treatment effectiveness [8]. Currently, a vast spectrum of therapeutic approaches is available to treat urological cancers. Surgery, chemotherapy, and radiation therapy are the most used cancer treatments [9]. Other modalities, namely hormonal, targeted, and immune therapies, have emerged in recent years to improve or overcome the low specificity of traditional therapeutic approaches, demonstrating significant clinical benefits in patient outcomes [10]. However, the development of resistance after long-term exposure continues to jeopardize the efficacy of chemotherapy, hormone therapy, and targeted therapy [11,12,13], and only a small percentage of patients respond to immunotherapy [14,15], depending on the identification of predictive biomarkers of therapeutic response.

In this regard, personalized medicine seeks to identify the most appropriate treatment option for a patient, considering both its individual genetic and non-genetic characteristics, to guide more informed clinical decisions, reducing adverse effects and associated costs [8,16,17,18]. In personalized medicine, cancer patients must be divided into distinct subgroups so that the optimal/most precise treatment can be prescribed at the right time [19]. As a result, the conventional “one treatment fits all” paradigm, in which treatment is used in a universal population approach regardless of individual genes, environments, and lifestyles, is changing [20]. The use of reliable cancer classifiers can help clinicians to correctly stratify patients to predict their therapeutic responses. Hence, pharmacogenomic studies have been conducted to investigate the influence of inter and intra-patient genetic variabilities on drug response [21]. However, pharmacogenomics does not consider other important factors that may affect therapeutic responses (e.g., gender, nutrition, age, health status, among others) [22]. These factors, as well as the disease state, can alter the metabolic profile (metabolome) of an individual, making metabolome analysis during a treatment regimen a promising strategy for identifying biomarkers of therapeutic response. Thus, pharmacometabolomics emerges to perform metabolic profiling of an individual prior, during, and after a treatment to identify potential metabolic biomarkers that can predict patient treatment response, which is of utmost interest in precision medicine [22,23]. This review focuses on the pharmacometabolomic studies that have been conducted over the last twelve years (2010 to 2021) to investigate biomarkers of therapeutic response in the most incident urological cancers, namely PCa, BCa, and RCC.

## 2. Current Status and Limitations of Therapies for Urological Cancers

The standard management of urological cancers initiates with surgery. However, alternative options must be used to control cancer progression at the metastatic stages. In this section, the most used strategies in the treatment of each urologic cancer are reviewed, starting with the most prevalent—PCa, followed by BCa and RCC.

### 2.1. Prostate Cancer

PCa is a complex disease with a large spectrum of aggressiveness, from localized PCa, which corresponds to 77% of the diagnosed PCa cases, to advanced/metastatic disease [24]. For low risk of progression (localized PCa) cases, surgery is recommended (prostatectomy) besides surveillance to prevent the substantial associated risk to develop recurrences [25,26,27]. Radiation therapy can also be prescribed for low- and intermediate-risk cases and can be divided into two major types: external beam radiation and brachytherapy [25,28]. External beam radiation is addressed to the prostate gland using high-energy rays from a machine outside the body. In turn, brachytherapy uses small radioactive particles, also called pellets or “seeds”, that are put directly into the prostate [28]. However, PCa can spread limiting its management by surgery or radiation and, therefore, patients can experiment relapses. In these cases, androgen deprivation therapy (ADT), also known as hormone therapy, is considered [29]. Male hormones, namely testosterone, are essential for prostate cancer development, maintenance, and progression [30]. ADT decreases the concentration of androgens in body circulation using gonadotropin-releasing hormones (GnRH) agonists (e.g., leuprolide, goserelin, triptorelin, histrelin) and antagonists (e.g., degarelix) as well as androgen inhibitors (e.g., bicalutamide, nilutamide, flutamide, enzalutamide) [29,30]. The most used hormone therapy drugs are listed in Table 1. Despite the benefits, several adverse effects have been associated with ADT including sexual side effects (e.g., loss of libido, erectile dysfunction, among others), osteoporosis, anemia, fatigue, depression, and cardiovascular complications [31]. Moreover, after an initial response to ADT, some patients develop castration-resistant PCa (CRPCa), a progression of the disease [32]. Novel anti-androgen drugs such as bicalutamide, nilutamide, or enzalutamide have been developed to treat this condition but the prognosis of CRPCa remains poor [25,32]. In these cases, ADT is commonly recommended for symptom control or prescribed in combination with chemotherapy (e.g., docetaxel, cabazitaxel, mitoxantrone) to slow the progression of advanced PCa [28,30]. Additionally, pharmaceutical compounds like olaparib and rucaparib were recently approved by the FDA for targeted therapy in patients with metastatic or hormone-resistant PCa who have mutations in breast cancer genes (BRACs), which are well-established tumor suppressor genes that maintain genomic stability [33,34].

The use of diagnostic biomarkers, such as the serum prostate-specific antigen (PSA), which is commonly used for PCa screening, prostatic acid phosphate (PAP), and prostate-specific membrane antigen (PSMA), as target antigens in immunotherapy has been investigated [32,35]. Moreover, the scientific community has been interested in combining immunotherapy and targeted therapy to increase the sensitivity of cancer cells to antitumor effects of antineoplastic drugs [36].

Apart from efforts to improve PCa therapeutics, reliable biomarkers for patient stratification and outcome prediction are urgently needed [32].

### 2.2. Bladder Cancer

Transitional cell carcinoma, which represents 90% of all BCa cases, presents distinct histological variants associated with different clinical features and outcomes [37]. Management of BCa is based on the pathological findings of the biopsy (histology, grade, and invasion), accomplished during the cystoscopy [38]. Local disease, also known as non-muscle-invasive BCa (NMIBC), is treated with transurethral resection of the bladder tumor (TURBT). However, approximately 20% of all NMIBC cases progress to muscle-invasive BCa (MIBC) which treatment is accomplished with radical cystectomy with pelvic node dissection [39]. Additionally, surgery is frequently combined with cisplatin-based neoadjuvant chemotherapy, which has been considered the first-line treatment for metastatic BCa [38,40]. Cisplatin-based treatment encompasses significant benefits for the patient, but it also causes several side effects and toxicity. Furthermore, it has a low response rate (40–50%) due to the development of resistance mechanisms during long-term exposures [41,42].

To address the drawbacks of chemotherapy, recent research has demonstrated the antitumoral activity of immune checkpoint inhibitors such as pembrolizumab and atezolizumab in BCa management [42]. These pharmaceutical compounds are currently recommended as first-line treatment for advanced BCa and non-responding or non-eligible chemotherapy BCa patients [39,43].

Mutations on FGFR3 or FGFR2 genes are usually associated with the presence of cancer. Indeed, FGFR3 has been used as a prognostic and predictive marker for less aggressive forms of BCa, as well as a therapeutic target of erdafitinib, which is typically recommended for cases of recurrent BCa [41,44]. These findings enhance the clinical benefits of identifying driving changes in the development of patient-specific therapies. In this vein, the possibility of predicting sensitivity to phosphatidylinositol 3-kinase/mammalian target of rapamycin (PIK3/mTOR) inhibitors has already been reported through the identification of mutations in the PIK3CA genes, that occur in approximately 26% of BCa patients [45].

Despite the advances in BCa management, response to treatment is still non-durable and cisplatin therapy is associated with significant treatment-related toxicity. Strategies for optimization of drug doses and drug combinations must be established in order to avoid drug overexposure. Moreover, the identification of early signs of resistance can help researchers gain a better understanding of the mechanisms that underpin resistance development.

### 2.3. Renal Cell Carcinoma

RCC is recognized as a metabolic disease characterized by deregulations on several processes including angiogenesis (e.g., von Hippel–Lindau/hypoxia-inducible factor gene mutations) [46], energy metabolism (Warburg effect), and nutrient-sensing pathways [47,48]. This heterogeneity impairs the selection of RCC therapeutics, hence, the definition of prognostic and predictive elements of therapeutic response is of utmost importance. Approximately 65% of all RCC diagnosed cases are confined to the kidney and can be successfully managed with surgery (total or partial nephrectomy), with a 5-year relative survival rate of 93% [49,50]. Despite the recognition of surgery as the standard treatment for localized RCC, in some particular cases, radiotherapy can be considered using a radiation source outside the body [7,51]. For metastatic RCC patients, the survival rate decreases significantly to 70% when cancer spreads to near structures of the kidney (e.g., lymph nodes) and 12% when cancer spreads to distant parts of the body [49]. In these cases, and when patients experience relapses after local therapy, systemic treatment is required. The systemic treatment begins with first-line therapies that include inhibitors of tyrosine kinases (e.g., vascular endothelial growth factor (VEGF) receptors), as sunitinib, pazopanib, tivozanib and cabozantinib, and mTOR, as temsirolimus and everolimus [7,52]. More recently, the combination of these targeted drugs with immunotherapeutic agents (ipilimumab/nivolumab or pembrolizumab/axitinib) has been approved [53,54]. Resistance acquisition is an important phenomenon in RCC in both targeted and immune therapies that contributes to limited long-term responses and, consequently, a poor prognosis [55]. The discovery of biomarkers that predict drug efficacy can help to avoid these problems, reducing costs and improving patient survival [56]. Ongoing and future clinical trials must delve deeper into the subset of patients who do not respond to current therapies. Chemotherapy is not recognized as a usual treatment for RCC due to low associated response rates [57].

## 3. Pharmacometabolomics

### 3.1. The Concept

Molecular profiling of tumors using genomics, proteomics, and metabolomics approaches can assist clinicians in patient stratification [8]. Pharmacogenomics, broadly defined as the study of individual genetic profiles to predict the therapeutic response of patients, has been used as a route to perform precision medicine but it is not sensitive to environmental factors (e.g., diet, age, nutrition, gender, lifestyle, gut microbiome among others) that also influence the response of a patient or a group of patients to the drug therapy [58]. Thus, pharmacometabolomics was defined in 2006 as “*an enhanced understanding of mechanisms for drug or xenobiotic effect and increased ability to predict individual variation in drug response phenotypes, based on using both baseline metabolic profiles prior to treatment and also effects of drug treatment over time*” [59]. Since metabolic changes precede phenotypic changes, pharmacometabolomics is a powerful tool to predict therapeutic responses based on the discovery of similar metabolic signatures of a group of patients pre- and post-treatment (metabotypes) [22,59,60]. These metabolic signatures refer to several low-molecular-weight endogenous and exogenous metabolites including amino acids, fatty acids, organic acids, carbohydrates, among others, and their levels can be used to build models of patient response to a drug (including toxicity and side effects) [61]. In the realm of precision medicine, the analysis of perturbations in the levels of these chemical compounds is critical to select biomarkers capable of predicting responses and monitoring the health status of a patient during treatment [62].

### 3.2. Workflow

The design of a pharmacometabolomic study constitutes an important point before its beginning to ensure optimal sample handling, metabolite extraction, detection, and interpretation to guarantee the reproducibility of the results [23]. Since the primary goal of pharmacometabolomics in the biomarker discovery field relies on the assumption that the metabolic fingerprint is altered by treatment response, resistance, and/or toxicity, the workflow initiates with the collection of biological samples before (baseline) and after treatment (Figure 1). The second step in the pharmacometabolomics pipeline is data acquisition which includes metabolite detection by high-throughput analytical techniques. Data generated in the second step must be processed into a manageable format using bioinformatic tools before being subjected to multivariate and univariate statistical methods. The final step entails data interpretation to identify potential biomarkers of therapeutic response (e.g., biomarkers of efficacy, toxicity, pharmacokinetics, and pharmacodynamics) to build mathematical models capable of predicting therapeutic responses.

The biological matrices most used in pharmacometabolomic studies include blood plasma/serum, tissues, and cells and they are collected prior to treatment at the baseline, followed by collection during and after treatment. Serum and plasma are complex matrices that are broadly used to monitor drug responses and can be collected at different time points (e.g., routine clinical follow-ups) [63]. Tissue samples require an invasive collection, and the representativeness of the tissue sample is compromised by the heterogeneity of the tumor; notwithstanding, they allow for the evaluation of the metabolites at their origin, i.e., in the specific organ [64]. For urological cancer studies, tissue can be collected during surgeries (e.g., prostatectomy, nephrectomy, and cystectomy), which are the primary treatment modality for most of them, in order to address implications of a previous drug exposure (e.g., adjuvant chemotherapy). However, recruiting a significant number of participants to collect human samples remains a challenge in pharmacometabolomic studies, making it necessary to investigate other biological models. Indeed, some studies referred to the use of biological samples from less complex models, such as *in vitro* and animal models including the cell line-derived xenograft (CDX), for accessing *in vivo* therapeutic responses. *In vitro* models provide an easier comprehension of drugs action and aid in their rapid incorporation into novel therapeutic settings [65,66]. Patient-derived xenografts (PDX) are models in which the tissue or cells from a patient’s tumor are implanted into an animal model (typically mice), allowing researchers to track the dynamics and progression of cancer (e.g., development of treatment resistance) while preserving tumor heterogeneity, genomic, and histological characteristics [67].

Regarding the analytical techniques used in pharmacometabolomics studies, nuclear magnetic resonance spectroscope (NMR) and mass spectrometry-based (MS) have been widely used in metabolomics studies [68]. Each one presents specific advantages and limitations in terms of sensitivity, reproducibility, and equipment costs [69]. No single analytical technique can provide complete coverage of the human metabolome which is composed of numerous metabolites from different classes with varying concentrations and physicochemical properties [68]. As a result, the combination of both NMR and MS-based platforms allows for a more accurate characterization of the physiologic, pathologic, and treatment-specific metabolites [70,71]. Despite this, the majority of pharmacometabolomic studies are limited to a single technique.

Translating the complex and non-linear data generated by NMR and MS methods to a model that can predict patient response to a drug is challenging. Thus, data pre-processing is a paramount step to prepare the raw data into a format that can be interpreted by multivariate analysis [72]. In general, the basic tools for data pre-processing include noise filtering and baseline correction, peak detection and deconvolution, alignment, and normalization [73]. Then, appropriate unsupervised and supervised multivariate analysis can ensure the extraction of useful information from experimental data [74,75]. Principal component analysis (PCA) is an unsupervised method to easily detect trends and outliers among samples, without considering sample identity. Hierarchical cluster analysis is also commonly used to build a tree diagram based on the similarity or differences between sample groups (clusters). Supervised methods, such as partial least squares-discriminant analysis (PLS-DA) and the orthogonal PLS-DA (OPLS-DA), consider a previous classification of samples to build the projection, maximizing the differences between samples [72]. Due to the risk of overfitting, model robustness must be evaluated through cross-validation or permutation tests and, if possible, using an independent set of data to build the models [72]. Univariate analysis (e.g., ANOVA, *t*-tests) is commonly used to evaluate the statistical significance of the biomarkers. To avoid the chance of false-positive results, *p*-values can be corrected for multiple comparisons using Bonferroni correction or the false discovery rate [72].

Discriminant metabolites identified can be further investigated to understand derangements in pathways that underlie drug response [22]. Several integration databases for enrichment analysis are available (e.g., MetaboAnalyst [76], Kyoto Encyclopedia of Genes and Genomes (KEGG) [77]) that can simplify the biochemical interpretation and relevance of the potential biomarkers [72,75].

## 4. Pharmacometabolomic Studies in Urological Cancers

The goal of pharmacometabolomic studies performed in the urological cancers field is to identify discriminative metabolites between samples collected before and after a drug treatment to advance with candidate biomarkers and mechanistic pathways of treatment response. In this section, a review of the literature was performed to access studies that apply pharmacometabolomic approaches to identify the metabolic responses to hormone therapy, immunotherapy, targeted therapy, and chemotherapy in PCa, BCa, and RCC. The identification of papers was conducted through a search on the PubMed database considering the following keywords or expressions [(metabolomics OR “metabolic profiling”) AND (chemotherapy OR “targeted therapy” OR immunotherapy OR “hormone therapy” OR “endocrine therapy” OR “androgen deprivation therapy” OR resistance) AND (“prostate cancer” OR “bladder cancer” OR “renal cancer” OR “kidney cancer” OR “renal cell carcinoma”)]. The search was performed in December 2021, considering all literature published in English from 2010 to 2021. In total, the search retrieved 163 papers (Figure 2). After an initial screening, 59 review papers and comments were excluded. The remaining 104 records of original results were analyzed, and 68 articles were excluded due to the lack of relevance for the topic, 10 articles reporting results with alternative pharmaceutical compounds, and 5 articles using other omics approaches. Finally, 18 articles were considered for the present review.

### 4.1. Pharmacometabolomic Studies in Prostate Cancer

PCa is the most studied urological cancer using pharmacometabolomic approaches with ten studies reported from 2010 to 2021. Table 1 summarizes the study design and main findings from those reports, including the pattern of metabolites with the potential to be candidate biomarkers of therapeutic response and the related metabolic pathways. Because castration resistance is a common condition in PCa, specific biomarkers associated with resistance development are also discussed in the research studies. Early biomarkers of resistance must be identified to optimize treatment regimens and improve long-term outcomes.

Three studies aiming to monitor the treatment response of PCa cells were performed using *in vitro* models [78,79,80]. Lodi et al. [78] used ^1^H NMR spectroscopy to assess the effects of treatment with the phosphatidylinositol 3-kinase (PI3K) inhibitor LY294002 and the heat shock protein 90 (HSP90) inhibitor 17AAG on the metabolome of two PCa cell lines. These compounds belong to the group of targeted therapy approaches, whose clinical interest has increased in the last few years. The pharmacometabolomic analysis was complemented with Western blotting to confirm the inhibition of the target proteins. The results showed that LY294002 treatment increased intracellular levels of glutamine and several branched-chain amino acids (BCAAs), such as valine, leucine, and isoleucine, while decreasing levels of lactate, alanine, fumarate, phosphocholine, and glutathione were observed. Regarding 17AAG, PCa-exposed cells demonstrated a similar increasing tendency in the levels of BCAAs, as well as increases in phosphocholine, *myo*-inositol, taurine, and citrate, and decreases in lactate, alanine, fumarate, and glutamine levels. Both inhibitors induced intracellular metabolic alterations (lactate, alanine, and fumarate) that may be associated with the activation of glucose uptake and glycolysis by PI3K-protein kinase B/Akt (PI3K/Akt) signaling. The alterations in intracellular glutamine levels unveiled some specificity for the treatment type, with an increase following PI3K inhibition (LY294002) and a decrease following HSP90 inhibition (17AA treatment), indicating a potential role of glutaminolysis in cancer cell growth. Citrate is typically found in high concentrations in healthy prostate tissue due to the specific metabolism of prostate cells that accumulates this metabolite rather than oxidizing it for energy production [71]. For this reason, the increase in intracellular concentration of citrate after 17AAG treatment may indicate a specific shift to a more physiological behavior of PCa cells. Furthermore, the process of obtaining energy is switched to glycolysis rather than oxidative phosphorylation in PCa, resulting in an increase in the levels of glucose and lactate compared to normal cells [71,81]. Given this, the decreased levels of lactate in PCa cells after treatment, found by Lodi et al. [78], suggested its effectiveness. Because tumors rely on BCAA intake for energy, their intracellular accumulation appears to indicate a slowdown of protein synthesis and, consequently, a decrease in cell proliferation and cancer progression [78,82]. The alterations found in common in both cell lines suggested that several metabolites are simultaneously modulated following treatment.

The *in vitro* study performed by Qu et al. [79] aimed to study the antitumor effects of proxalutamide and two others currently used androgen receptor (AR) antagonists (bicalutamide and enzalutamide) on AR-positive (LNCaP and 22RV1) and AR-negative PCa cell lines through liquid chromatography quadrupole time-of-flight mass spectrometry (LC-Q/TOF-MS). The results unveiled that proxalutamide significantly decreased the intracellular levels of glutamine, glutamate, cysteine, glycine, reduced glutathione (GSH), oxidized glutathione (GSSG), aspartate, uridine, cytidine, and thymidine in AR-positive cells, while no effect was observed on the intracellular levels of AR-negative cell lines. These changes suggested that proxalutamide inhibited glutamine metabolism, redox homeostasis (glutathione metabolism), and pyrimidine synthesis in AR-positive cells. Moreover, a reduction in the expression of cell surface transporters necessary for glutamine intake was noted through Western blot assays.

In the third study, the metabolic dysregulations in castration-resistant PCa compared to androgen-dependent PCa were investigated in *in vitro* and animal models by targeted NMR spectroscopy analysis of metabolites involved in the metabolism of [U-^13^C]-glucose and [U-^13^C]-glutamine [80]. The *in vitro* model comprised the intracellular metabolites of human PCa cell lines representative of androgen-dependent PCa (LNCaP) and castration-resistant PCa (PC3), which were cultured with ^13^C-glucose and ^13^C-glutamine media. In the animal model, aqueous extracts of solid tumor masses from the transgenic adenocarcinoma of mouse prostate (TRAMP) mice were analyzed. The results unveiled higher levels of amino acids, membrane precursors, and organic acids from energy metabolism in the resistant condition, indicating a higher energetic and biosynthetic demand to support cell survival and growth. Interestingly, this study reported an increase in the levels of glutamate (an intermediate in glutathione metabolism) as well as glutathione, an antioxidant agent that can protect resistant cells from the cytotoxic effects of drugs, thereby supporting the development of drug resistance [80,83].

**Table 1 pharmaceuticals-15-00295-t001:** Pharmacometabolomic studies performed in PCa.

Cancer Therapy under Study	Samples	Instrumental and Statistical Analysis	Treatment Response	Metabolic Interpretation	Ref.
***In vitro* and animal models**					
Targeted therapy: PI3K inhibitor LY294002 (10–25 μM) and HSP90 inhibitor 17AAG (0.25–1 μM) 48 h of treatment exposure	Intracellular (polar) metabolome of PCa cell lines: PC3 untreated (DMSO solvent control), *n* = 8 treated with LY294002, *n* = 8 treated with 17AAG, *n* = 8 LNCaP untreated (DMSO solvent control), *n* = 8 treated with LY294002, *n* = 8 treated with 17AAG, *n* = 8	^1^H NMR PCA Mann–Whitney U test	LY294002 treatment effects in both cell lines (PC3 and LNCaP): ↑ valine; leucine; isoleucine; glutamine ↓ alanine; lactate; fumarate;glutathione; phosphocholine 17AAG treatment effects in both cell lines (PC3 and LNCaP): ↑ valine; leucine; isoleucine; phosphocholine; *myo*-inositol; taurine; citrate ↓ lactate; alanine; fumarate; glutamine	LY294002 and 17AAG exposure activated glycolysis by PI3K/Akt signaling and influenced the glutaminolysis	[78]
Hormone therapy: androgen receptor (AR) antagonists proxalutamide, bicalutamide, and enzalutamide (1–10 μM) 48 h of treatment exposure	Intracellular (polar) metabolome of PCa cell lines: AR-positive cells (22RV1 and LNCaP): untreated, *n* = 6 treated with each AR antagonist, *n* = 6 per drug AR-negative cells (PC3, DU145): untreated, *n* = 6 treated with AR antagonist, *n* = 6 per drug	LC-Q/TOF-MS PCA, PLS-DA, OPLS-DA Two-tailed student’s *t*-test One-way analysis of variance	Proxalutamide treatment effects in both AR-positive cell lines: ↓ glutamine; glutamate; GSH; GSSG; GSH/GSSG; glycine; aspartate; uridine, cytidine; thymidine Bicalutamide treatment effects in AR-positive cell lines: ↓ thymidine Enzalutamide treatment effects in AR-positive cell lines: ↓ GSH ↑ aspartate No significant changes were found for proxalutamide, bicalutamide, and enzalutamide in AR-negative cell lines	Proxalutamide exposure inhibited glutamine metabolism, glutathione metabolism and pyrimidine metabolism	[79]
Hormone therapy	Intracellular (polar) metabolome of cell lines: AR-positive cells: LNCaP, *n* = 4 Castration resistant cells: PC3, *n* = 4 Tissue extract (polar phase): TRAMP, *n* = 3 castrate resistant TRAMP, *n* = 3	^1^H NMR Student’s *t*-test	Castration resistant condition effects in cell lines: ↑ aspartate; glutamate; lactate; *myo*-inositol; phosphocholine; glycerophosphocholine; total choline; alanine; glutathione ↓ citrate; glucose; creatine; creatine phosphate Castration resistant condition effects in TRAMP: ↑lactate; aspartate; glutamate; glutathione ↓citrate; creatine	Castration resistant condition was associated with an upregulation of glycolysis; TCA cycle; glutaminolysis and glutathione synthesis	[80]
**Human models**					
Hormone therapy: leuprolide (22.5 mg IM 3-month depot) and bicalutamide (50 mg per day) 4 weeks of treatment exposure	Lipophilic and hydrophilic plasma extracts: PCa untreated group, *n* = 36 PCa treated group, *n* = 36	LC-MS/MS GC-MS Student’s *t*-test	Hormone therapy effects in PCa treated group: ↑cholate ↓dehydroisoandrosterone sulfate; epiandrosterone sulfate; androsterone sulfate; cortisol; 4-androsten-3β, 17β-diol disulfates 1 & 2; 5α-androstan-3β,17β-diol disulfate; pregnendiol disulfate; pregn steroid monosulfate; andro steroid monosulfates 1; deoxycarnitine; acetylcarnitine; hexanoylcarnitine; octanoylcarnitine; decanoylcarnitine; laurylcarnitine; palmitoylcarnitine; stearoylcarnitine; oleoylcarnitine; 3-hydroxybutyrate; acetoacetate; dodecanedioate; octadecanedioate; 2-hydroxybutyrate; α-hydroxyisovalerate; 2- methylbutyroylcarnitine	Hormone therapy exposure inhibited steroids synthesis, fatty acid oxidation, bile acid synthesis and BCAAs synthesis	[84]
Hormone therapy: bicalutamide and goserelin up to 2 years of treatment exposure	Lipophilic serum extract: PCa untreated group, *n* = 18 treated group, *n* = 36 (poor response *n* = 18 and good response *n* = 18) Healthy group, *n* = 18	LC-MS PLS-DA OPLS Duncan pairwise post hoc tests	Hormone therapy effects in PCa poor response group: ↑ deoxycholic acid; glycochenodeoxycholate; L-tryptophan; arachidonic acid; deoxycytidine triphosphate; pyridinoline ↓ docosapentaenoic acid Hormone therapy effects in PCa good response group: ↑ L-tryptophan; arachidonic acid; deoxycholic acid; glycochenodeoxycholate ↓ docosapentaenoic acid; pyridinoline; deoxycytidine triphosphate	Hormone therapy exposure altered cholesterol metabolic pathway	[85]
Hormone therapy: degarelix (240 mg) 7 days of treatment exposure	Tissue extract (polar phase): PCa untreated group, *n* = 6 treated group, *n* = 7 Control group, *n* = 10	^1^H-NMR PCAOPLS-DA	Degarelix effects in PCa treated group: ↓lactate; total choline	Hormone therapy exposure reduced glycolysis and membrane phospholipid metabolism	[86]
Hormone therapy: LHRH agonist, LHRH-antagonist, or orchiectomy 3 and 6 months of treatment exposure	Lipophilic and hydrophilic serum extracts: PCa untreated group, *n* = 20 treated group (3 months), *n* = 20 treated group (6 months), *n* = 20	GC-TOF-MS LC-HILIC-MS/MS Volcano plot ANOVA Pearson correlation	Hormone therapy effects in PCa treated group (3 months): ↑ dihydroxycholestanoyl taurine; dodecanedioic acid; eicosatetraenoic acid ↓ hydroxymyristoyl-carnitine; malonyl-carnitine; hexanoyl-carnitine; dodecenoyl-carnitine; octanoyl-carnitine; oleoyl-carnitine; decanoyl-carnitines; 3-hydroxyburtirc acid; indoleacetic acid; andosterone sulfate Hormone therapy effects in PCa treated group (6 months): ↑ dihydroxycholestanoyl taurine; AMP; N-acetyl-glucosamine-1-phosphate; mevalonate-5-phosphate; 2-hydro-D-gluconate ↓ malonylcarnitine; oleoylcarnitine; hexanoylcarnitine; tetradecendoycarnitine; heptanoylcarnitine palmitoylcarnitine; decanoyl-carnitines; myristolylcarnitine; 3-hydroxyburtic acid; oxalic acid; glycolic acid; nonanoic acid; androsterone sulfate	Hormone therapy exposure reduced steroid biosynthesis; fatty acid β-oxidation and ketogenesis and alters microbiome metabolism	[87]
Hormone therapy	Tissue extract (polar phase): BPH group, *n* = 39 HSPCa group, *n* = 39 CRPCa group, *n* = 25	^1^H NMR PCA, OPLS-DA 10-Fold cross validation CV-ANOVA Student *t*-test Bonferroni correction AUC	Hormone therapy resistance effects in CRPCa (compared with BPH): ↑ alanine; lactate; glutamate; taurine ↓ *myo*-inositol; citrate Hormone therapy resistance effects in CRPCa (compared with HSPV): ↑ creatine ↓ choline; lactate; alanine; glutamate; glycine	Castration resistant condition was associated with down-regulation of amino acid metabolism; membrane metabolism (choline metabolism) and altered energy metabolism (possibility of inverse Warburg effect)	[88]
Chemotherapy: docetaxel (75 mg/m^2^) 3 weeks of treatment exposure	Lipophilic plasma extract: Discovery set: PCa untreated group, *n* = 96 treated group, *n* = 89 Validation set: PCa untreated group, *n* = 63 treated group, *n* = 47	LC-MS/MS Latent class analysis Univariable and multivariable cox regression Logistic regression Student’s *t*-test	Docetaxel effects in PCa treated group: no significant changes were found	-	[89]
Chemotherapy (docetaxel) and hormone therapy (LHRH analog) 18–24 weeks of treatment exposure	Tissue extract (polar phase): PCa treated group, *n* = 12 untreated group, *n* = 10	HPLC PCA OPLS-DA Two-tailed student’s *t* test One-way analysis of variance	Docetaxel and hormone therapy effects in PCa treated group: ↑ GSSG; glycerol-3-phosphate; shikimate; 14.0 Lyso PA; d-glucarate; dodecylbenzenesulfonic acid; guanosine ↓ phospholipids (PC, PE, PS, LPE); 27-hydrocycholesterol 3 sulfate; 2-hydroxy-4-methylpentanoate	Docetaxel and hormone therapy exposure inhibited pathways involved in biosynthesis and energy metabolism: amino acid metabolism; purine and pyrimidine metabolism; TCA cycle; lipid synthesis; glutathione metabolism	[90]

22RV1: human prostate carcinoma epithelial cell line; AMP: adenosine monophosphate; AUC: area under curve; AR: androgen receptor; BCAAs: branched chain amino acids; BPH: benign prostatic hyperplasia; CRPC: castration resistant PCa; CV-ANOVA: analysis of variance of cross-validated residuals; DU145: cell line with epithelial morphology isolated from the prostate; GC-MS: gas chromatography–mass spectrometry; GC-TOF-MS: gas chromatography-time of flight-mass spectrometry; GSH: reduced glutathione; GSSG: oxidized glutathione; HPLC: high-performance liquid chromatography; ^1^H-NMR: proton nuclear magnetic resonance spectroscopy; HSP90: heat shock protein 90; HSPC: hormone sensitive PCa; LC-MS/MS: liquid chromatography-tandem mass spectrometry; LC-HILIC-MS/MS: liquid chromatography-hydrophilic interaction chromatography-tandem mass spectrometry; LC-HRMS: liquid chromatography-high resolution mass spectrometry; LC-MS: liquid chromatography-mass spectrometry; LNCaP: lymph node carcinoma of the prostate; LPE: lysophosphatidylethanolamine; PA: phosphatidic acid; PC3: prostate cancer cell line; PC: phosphatidylcholine; PCa: prostate cancer; PCA: principal component analysis; PE: phosphatidylethanolamine; PI3K: phosphatidylinositol 3-kinase; PS: phosphatidylserine; OPLS-DA: orthogonal partial least squares-discriminant analysis; TRAMP: transgenic adenocarcinoma of mouse prostate; UMP: uridine monophosphate.

Human samples have been the preferred model for pharmacometabolomic studies of PCa, but the number of available samples collected before and after treatments has been limited. Table 1 summarizes seven pharmacometabolomic studies performed in serum, plasma, and tissue from PCa patients under hormone therapy (androgen deprivation therapy) [84,85,86,87,88] and chemotherapy regimens [89,90]. The first study was performed by Saylor et al. [84] to investigate changes in plasmatic metabolite levels in patients receiving a gonadotropin-releasing hormone (GnRH) agonist using both GC-MS and LC-MS/MS. The main alterations observed in plasma extracts after 3 months of treatment exposure included an increase in the levels of cholate (an intermediate of bile acid metabolism) and a decrease in the levels of steroids and metabolites from lipid β-oxidation (several carnitines and ketone bodies), effects which were associated with the use of androgen-deprivation therapy in PCa patients [84,91].

One of the most recent studies presented in Table 1 was performed by Chi et al. [87] and corroborated the findings reported in the aforementioned study (Saylor et al. [84]). Chi et al. carried out a pharmacometabolomic study with the goal of investigating serum metabolic alterations in PCa patients after hormone therapy at three different time points: before, 3 months, and 6 months after therapy. The most significant changes were found consistently at 3 and 6 months and included significantly lower levels of androsterone sulfate, 3-hydroxybutyric acid, and several long-chain acyl-carnitines, suggesting an impact in steroid synthesis and ketogenesis. Furthermore, some of the known adverse effects of hormone therapy included glucose intolerance and insulin resistance [31,87].

The study carried out by Huang et al. [85] used LC-MS to analyze serum samples from PCa patients with different responses, i.e., good and poor responders, to the combination of bicalutamide and goserelin, which correspond to an antiandrogen and a luteinizing hormone-releasing hormone analog, respectively. Several metabolites were statistically altered between untreated PCa patients and healthy controls, including oxycholic acid, glycochenodeoxycholate, L-tryptophan, docosapentaenoic acid, arachidonic acid, deoxycytidine triphosphate, and pyridinoline. After drug treatment, these metabolites returned to near-normal levels in the good responder patient group, but not in patients who developed castration resistance (poor responders). Additionally, higher levels of docosapentaenoic acid and glycochenodeoxycholate were linked to a faster rate of cholesterol synthesis. These findings may be useful in predicting how patients will respond to this type of hormone therapy combination.

Regarding tissue analysis, Madhu et al. [86] investigated the metabolic effects of degarelix, a hormone blocker, as adjuvant treatment in PCa patients undergoing radical prostatectomy. Polar metabolites were extracted from benign and malignant tissue samples and analyzed by ^1^H-NMR spectroscopy. Degarelix treatment induced a significant decrease in the levels of lactate (fuel for oxidative metabolism) and total choline (membrane precursor). These metabolites are usually found to be elevated in PCa human samples [71], so their levels can be monitored and proposed as candidate biomarkers of therapeutic response to degarelix.

A better understanding of the metabolic differences between hormone-sensitive (HSPCa) and castration-resistant PCa (CRPCa) is also of utmost importance. In this way, recent research presented the ^1^H NMR analysis of tissue from patients with the two phenotypes under hormone therapy [88]. The results showed a good separation between the two groups, with choline, creatine, and lactate as the metabolites with the highest discriminatory power for the metabolic signatures of patients sensitive and resistant to hormone therapy. This potential biomarker panel suggested that the main alterations occurring in the resistant condition were related to alterations in amino acid metabolism, choline metabolism and suggested a possible inverse of the Warburg effect [88].

Lin et al. [89] used LC-MS/MS to investigate changes in plasma metabolites levels in PCa patients before (baseline) and after docetaxel (a chemotherapeutic agent) treatment to develop a tool for monitoring which patients should or should not remain with chemotherapy. However, the authors concluded that it was not possible to assign a metabolic signature that distinguishes between the two groups of patients. The authors justified the findings by pointing out important limitations in the study that may have contributed to the results; namely, the inability to identify the circulant lipid species and the small number of patients enrolled, which remains a limiting and dependent factor in studies using human samples. Notwithstanding, a recent study using the combination of docetaxel with a luteinizing hormone-releasing hormone analog (hormone therapy) before prostatectomy addressed the impact of this adjuvant combination on the metabolome of PCa tissue samples [90]. The results unveiled significant dysregulations in the levels of several metabolites including an increase in the levels of glycerol-3-phosphate, dodecylbenzenesulfonic acid, guanosine, among others, and a decrease in the levels of phospholipids, 27-hydroxycholesterol 3 sulfate, 2-hydroxy-4-methylpentanoate, among others. Since most of these metabolites belong to important pathways for cell growth and proliferation, such as biosynthesis and energy metabolism, the use of this type of combination before prostatectomy suggested a therapeutic benefit limiting tumor growth. Expression of key proteins in tumor samples from patients receiving or not receiving the neoadjuvant therapy was also evaluated to corroborate the findings obtained by HPLC analysis.

### 4.2. Pharmacometabolomic Studies in Bladder Cancer

Regarding BCa, Table 2 depicts the three pharmacometabolomic studies performed in the last 10 years including the methodology used and a summary of the main findings. Two *in vitro* pharmacometabolomic studies [13,92] were conducted to address the metabolic implications of cisplatin resistance and one additional study was performed to predict the response to gemcitabine as neoadjuvant therapy [93].

Although BCa tumors are initially susceptible to cisplatin, the development of resistance during treatment hampers its efficacy. In this regard, Lee et al. [13] performed a comparative lipidomic profiling of two BCa cell lines: a cisplatin-sensitive and a cisplatin-resistant cell line as a representation of *in vivo* chemoresistant BCa. Ultra-performance LC-MS (UPLC-MS) profiling of several lipid species revealed altered levels between the endometabolome of the two cell lines, including significantly elevated levels of one ceramide and two triglycerides in cisplatin-resistant cells. Ceramides are recognized as important elements in cellular membrane structure, whereas triglycerides are essential for energy storage [71], which reinforces the maintenance of cell growth to promote tumor progression on a resistance condition [13]. The authors concluded that a rearrangement of lipid metabolism is required for BCa pathogenesis.

The same in vitro model (resistant, T24R, and cisplatin sensitive cell lines, T24S) was explored by Wen et al. [92] that performed an NMR quantitative analysis of glucose-derived metabolites in cisplatin-resistant conditions. In comparison to sensitive cells, resistant cells consumed more glucose and consequently excreted more glucose-derivatives, acetate, and fatty acids. The authors also found a preference for glucose as a source of increased fatty acid synthesis in resistant cells by using labeled glucose. These findings agree with the previous study [13] confirming a reprogramming of lipid metabolism in cisplatin-resistant BCa.

The combination of gemcitabine and cisplatin is frequently recommended in BCa treatment. A study performed by Yang et al. [93] aimed to identify predictive biomarkers of efficacy to treatment with gemcitabine as neoadjuvant therapy for transurethral resection of bladder tumors. Metabolite analysis using high-resolution LC-MS was performed on tissue collected from BCa patients before and after submucosal gemcitabine injection. Adjacent healthy tissue was also collected for comparison with normal conditions. Tissue metabolic profiling unveiled significant alterations in the levels of several metabolites, most notably alterations on bilirubin and retinal, which recovered to near normal levels after gemcitabine treatment. The authors proposed that bilirubin and retinal could be investigated as therapeutic targets of gemcitabine.

### 4.3. Pharmacometabolomic Studies in Renal Cell Carcinoma

Sunitinib is commonly used for the treatment of advanced RCC, but 30% of patients are intrinsically resistant to this type of targeted agent [12]. In this regard, two studies were carried out with the goal of identifying biomarkers of resistance to sunitinib treatment using *in vitro* [83] and xenograft [94] models, and three others investigated the metabolic responses in human samples after exposure to different anti-neoplastic agents [95,96,97]. Therapeutic strategies for resistance situations are of utmost importance, particularly in the case of RCC, for which the systemic therapy effectiveness rates remain low. Table 3 lists five pharmacometabolomic studies focused on RCC therapeutics performed in the last 10 years, including the methodology used and a summary of the main findings.

The first study used capillary electrophoresis-time of flight mass spectrometry (CE-MS) to look at metabolic changes in sunitinib-resistant RCC cells [83]. The results revealed that intracellular levels of several metabolites involved in energy processes, including glycolysis, TCA cycle, and pentose phosphate pathway, were found to be higher in sunitinib-resistant cell lines [83]. Because glycolysis is the primary source of energy in this type of cancer cells, these mechanisms are essential for RCC progression [98]. These findings are consistent with a most recent study performed by Sato et al. [94] where the use of a xenograft RCC mouse model was reported to identify changes in intracellular metabolites unveiling the mechanisms of sunitinib resistance. In brief, they subcutaneously implanted both sensitive (786-P) and sunitinib-resistant (786-R) cell lines in different mice groups followed by administration of sunitinib. After 4 weeks of sunitinib exposure, tumoral tissues were collected from each mouse and subjected to primary cell culture. In the sunitinib-resistant condition, LC-MS analysis revealed significantly increased intracellular levels of fructose 6-phosphate, D-sedoheptulose 7-phosphate, and glucose 1-phosphate, suggesting a higher glycolysis rate together with a higher uptake of glutamine into the TCA cycle. These results were corroborated by the higher expression of glutamine transporters in sunitinib-resistant cells. Moreover, the results disclosed significantly higher levels of glutathione and *myo*-inositol in resistant primary cells compared to sensitive cells, indicating a higher antioxidant activity in resistant cells as a protective action against the antitumor effects of sunitinib. Future validation of the role of these pathways using sensitive and sunitinib-resistant RCC patient groups may identify early biomarkers of resistance.

The implications of sunitinib and other targeted therapies on the metabolism of patients with metastatic RCC were also investigated by Jobard et al. [96]. Pre-treatment and serial on-treatment serum samples were collected during a clinical trial and analyzed by NMR spectroscopy. The main goal of this study was to compare the metabolic response to two conventional treatments (sunitinib and interferon-α in combination with bevacizumab) with a new combination with bevacizumab, an anti-angiogenic drug, and temsirolimus, an mTOR inhibitor, proving the use of a pharmacometabolic approach to monitor and predict responses. The combination of bevacizumab and temsirolimus caused a faster and higher impact on the metabolome of treated RCC patients when compared to other treatments. The main dysregulated metabolites were lipid species and lipoproteins, which are metabolites responsible for the transport of endogenous lipids and cholesterol, metabolites of β-oxidation, as well as glucose, cholesterol, among others. These findings could be indicative of the associated side effects of this combination, as some toxic effects, such as hypercholesteremia and hypertriglyceridemia, have already been observed in RCC patients under temsirolimus treatment [99]. These results highlighted the potential of a pharmacometabolomics approach to predict chemotherapeutic side effects [96].

The pharmacometabolomic studies performed by Li et al. [97] and Mock et al. [95] investigated the impact of immunotherapy. The first study [97] used LC-MS to analyze serum samples collected from different RCC patients (at 4 and 8 weeks of treatment). The profiling of serum metabolites revealed an increase in the kynurenine/tryptophan ratio. Kynurenine is a product from tryptophan metabolism that causes immunosuppression. Tryptophan is usually found downregulated in RCC cells suggesting an increased use by these cells [100]. In this study, the alteration in the kynurenine/tryptophan ratio was associated with an adaptative resistance mechanism and, as a result, a worse overall survival of RCC patients. The monitoring of these metabolites can provide information about the patient’s condition during treatment.

Lastly, the study performed by Mock et al. [95] investigated the metabolic changes that underpin immunotherapy response and failure. To achieve this goal, serum samples from 28 urological cancer patients (25 RCC plus 2 BCa plus 1 patient diagnosed with both) who underwent immunotherapy were collected before the first, second, and third cycles, followed by LC-MS analysis. Most serum metabolites associated with response to immunotherapy belong to the class of very long-chain fatty acid-containing lipids (VLCFA-containing lipids). Comparing the serum lipid content between responders and non-responders among the first and third cycles, VLCFA-containing lipids seem to act as sensitizers to immune treatment, based on the impact of T cell metabolism, as suggested by the authors. The prognostic value of the VLCFA-containing lipids was also investigated by transcriptomic analysis. To confirm the potential of this metabolite class in predicting immunotherapy response, the obtained results must be validated in a larger patient cohort.

## 5. Conclusions and Future Perspectives

Despite the efforts made to date to improve cancer therapies, the selection of the best treatment modality for metastatic stages in urological cancers remains a challenge. Thus, the understanding of the impact of treatments on the metabolome of patients can be paramount for predicting therapeutic responses in the personalized medicine era. The studies published so far on this topic advanced with metabolic signatures characteristic of responses to hormone therapy and chemotherapy in PCa, along with chemotherapy in BCa and targeted therapy and immunotherapy in RCC. These metabolic signatures may lead to the definition of candidate predictive biomarkers of treatment response, as well as the recognition of metabolic alterations that occur during resistance development.

Further studies are needed to validate the impact of the different treatments on the metabolic fingerprint in larger and independent clinical cohorts of different populations (e.g., ethnicities) of urological cancers patients, as well as to complement those studies with multi-omics analyses to improve accuracy in predictive biomarker detection and a better understanding of the mechanisms underlying cancer therapeutic resistance. In this regard, a better stratification of patients based on their metabotypes (metabolic signatures) may help to predict its prognosis. In future pharmacometabolomics studies, it is important to consider the standardization of protocols of sample handling and storage, analytical conditions, and data interpretation. Human sample biobanks are an important aspect to consider in these studies because they preserve samples collected before and during a treatment regimen, increasing the availability of a higher number of human samples. Additionally, the incorporation of machine learning techniques can speed up the interpretation of big data generated in clinical practice by recognizing patterns of metabotypes to improve patient stratification and, as a result, devising optimal treatment strategies.

## Figures and Tables

**Figure 1 pharmaceuticals-15-00295-f001:**
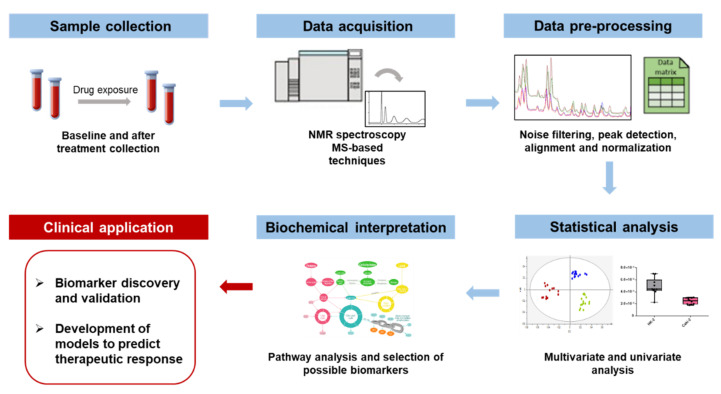
Overview of a representative pharmacometabolomics workflow and its main goals.

**Figure 2 pharmaceuticals-15-00295-f002:**
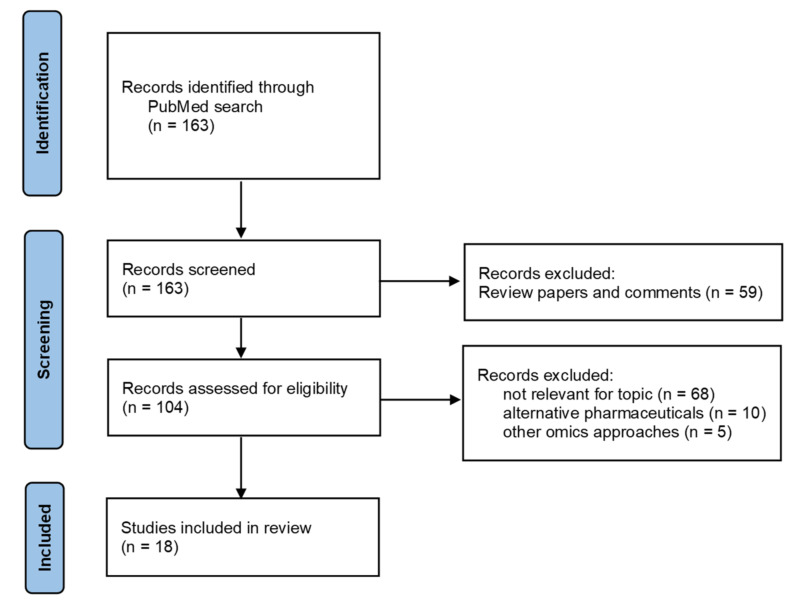
Flow diagram of literature search (time frame: 2010–2021; database: PubMed).

**Table 2 pharmaceuticals-15-00295-t002:** Pharmacometabolomic studies performed in BCa.

Cancer Therapy under Study	Samples	Instrumental and Statistical Analysis	Treatment Response	Metabolic Interpretation	Ref.
***In vitro* models**					
Chemotherapy: cisplatin (10 mM) 2 days of treatment exposure	Intracellular (lipophilic) metabolome of BCa cell lines: Cisplatin-sensitive cells T24S, *n* = 4 Cisplatin resistant cells T24R, *n* = 4	UPLC-MS PCA Student’s *t*-test *p*-value	Cisplatin resistance effects: ↑ CE (22:6); TG (49:1); TG (53:1)	Cisplatin-resistant condition altered lipid metabolism (storage of fatty acids and phospholipid biosynthesis)	[13]
Chemotherapy: cisplatin (10 μM) 12 h of treatment exposure	Intracellular (polar) metabolome of cell lines: Cisplatin-sensitive cells T24S, *n* ≥ 3 Cisplatin-resistant cells T24R, *n* ≥ 3	2D NMR (^1^H–^13^C HSQC) Student’s *t* test	Cisplatin resistance effects: ↑ acetate; fatty acids ↓ glucose; lactate; alanine:	Cisplatin-resistant condition was associated with an upregulation of glycolysis (Warburg effect) and fatty acid synthesis (cellular proliferation)	[92]
**Human models**					
Chemotherapy: gemcitabine (50 mg, dissolved in 20 mL normal saline) 30 min of treatment exposure	Lipophilic tissue extracts: BCa untreated group, *n* = 12 BCa treated group, *n* = 12 adjacent normal group, *n* = 12 adjacent normal treated group, *n* = 12	LC-HRMS PCA Paired student *t*-test	Gemcitabine effects in BCa treated group: ↓ bilirubin; retinal Gemcitabine effects in adjacent normal treated group: ↑ histamine ↓ thiamine	-	[93]

2D NMR: two-dimensional nuclear magnetic resonance; ^1^H–^13^C HSQC: ^1^H–^13^C heteronuclear single quantum coherence; BCa: bladder cancer. CE: ceramides; LC-MS/MS: liquid chromatography-tandem mass spectrometry; LC-HRMS: Liquid chromatography-high resolution mass spectrometry; LC-MS: liquid chromatography-mass spectrometry; PCA: principal component analysis; T24: human urinary bladder cancer patient cell line; TG: triglycerides; UPLC-MS: ultraperformance liquid chromatography-tandem mass spectrometry.

**Table 3 pharmaceuticals-15-00295-t003:** Pharmacometabolomic studies performed in RCC.

Cancer Therapy under Study	Samples	Instrumental and Statistical Analysis	Treatment Response	Biological Interpretation	Ref.
***In vitro* and animal models**					
Targeted therapy: sunitinib (10 mM) 5 days of treatment exposure	Intracellular (polar) metabolome of RCC cell lines: 786-O Par (parental), *n* = 3 786-O Res (sunitinib-resistant), *n* = 3	CE-TOF MS PCA Fold change Two-tailed student t-test	Sunitinib resistance effects: ↑ dihydroxyacetone phosphate; fructose 1,6-bisphosphate; choline; cysteine; methionine; thymidine; citrate; glycerophosphorate; fumarate; glucose-6-phosphate; tryptophan; ADP; creatine; 6-phosphogluconate; sedoheptulose-7-phosphate; fructose-6-phosphate; glutamate; malic acid; acetyl-CoA ↓ oxidized glutathione; ornithine; creatinine; guanine; succinic acid	Sunitinib resistant condition is associated with up-regulation on lipid biosynthesis (membrane metabolism), energy metabolism (glycolysis and TCA cycle), arginine and proline pathways, urea cycle and nucleic acid biosynthesis	[83]
Target therapy: sunitinib (25 mg/kg per day) 4 weeks of treatment exposure	Intracellular (lipophilic) metabolome of primary cell culture of xenograft RCC mouse model: 786-P (parental) untreated, *n* = 5 treated, *n* = 5 786-R (sunitinib-resistant) treated, *n* = 5	LC-MS/MS Mann–Whitney U test One-way ANOVA Post hoc Tukey’s test	Sunitinib resistance effects in 786-R (compared with 786-P treated): ↑ glutamine; 2-oxoglutaric acid; fructose 6-phosohate; D-sedoheptulose 7-phosphate; glucose 1-phosphate; myo-inositol Sunitinib resistance effects in 786-R (compared with 786-P untreated) 1-phosphate; fructose 6-phosohate; D-sedoheptulose 7-phosphate ↓ glutamate; glutathione; myo-inositol	Sunitinib resistant condition is associated with up-regulation of energy metabolism (glutamine uptake, glycolysis, and TCA cycle) and antioxidant activity	[94]
**Human models**					
Targeted therapy: Arm A- bevacizumab (10 mg/kg1 every 2 weeks) and temsirolimus (25 mg per week) combination Arm B—sunitinib (50 mg per day for 4 weeks followed by 2 weeks off) Arm C- interferon- α (9 mIU three times per week) and bevacizumab (10 mg/ kg every 2 weeks) combination 2 and 5–6 weeks of treatment exposure	Hydrophilic serum extracts: Arm A RCC untreated group, *n* = 56 treated group (2 weeks), *n* = 55 treated group (5–6 weeks), *n* = 49 Arm B RCC untreated group, *n* = 26 treated group (2 weeks), n = 22 treated group (5–6 weeks), *n* = 20 Arm C RCC untreated group, *n* = 20 treated group (2 weeks), *n* = 25 treated group (5–6 weeks), *n* = 22	^1^H NMR PCA OPLS Cross-validation ANOVA	Bevacizumab and temsirolimus combination effects in RCC treated group (2 weeks): ↑ glycerol backbone of phosphoglycerides; triacylglycerides; fatty acids; very low-density lipoproteins and low-density lipoproteins; glucose; N-acetylglycoproteins Bevacizumab and temsirolimus combination effects in RCC treated group (5–6 weeks): ↑ glycerol backbone of phosphoglycerides; triacylglycerides fatty acids; very low-density lipoproteins; low-density lipoproteins; glucose; N-acetylglycoproteins; BCAAs; alanine; glycine; glutamine; acetoacetate; acetone; glycerol; cholesterol ↓acetate; ethanol Sunitinib effects in RCC treated group (2 and 5–6 weeks): no significant changes were found Interferon-α and bevacizumab combination effects in RCC treated group (5–6 weeks): ↑ lipids and very low-density lipoproteins ↓ low-density lipoproteins	Bevacizumab and temsirolimus combination caused the greatest modification essentially in lipid and lipoprotein metabolisms	[96]
Immunotherapy: Arm A- nivolumab (phase 1: 0.3/3/10 mg/kg every 3 weeks; phase 3: 3 mg/kg every 2weeks) Arm B- everolimus phase 3 (10 mg per day) 4 and 8 weeks of treatment exposure	Lipophilic serum extracts: Arm A Phase 1 trial RCC untreated group, *n* = 91 treated group (4 weeks), *n* = 84 treated group (9 weeks), *n* = 69 Phase 3 trial RCC untreated group, *n* = 392 treated group (4 weeks), *n* = 98 treated group (8 weeks), *n* = 324 Arm B RCC untreated group, *n* = 349 treated group (4 weeks), *n* = 58 treated group (8 weeks), *n* = 0	LC-MS Volcano plots Benjamin-Hochberg multiple testing corrections Pearson correlations	Nivolumab effects in RCC treated group (phase 1 and 3): ↑ kynurenine Everolimus effects in RCC treated group: no significant changes were found	Nivolumab exposure upregulated tryptophan catabolism (increased tryptophan to kynurenine conversion) resulting in an adaptive immune suppressive microenvironment	[97]
Immunotherapy (checkpoint inhibitors): nivolumab and atezolizumab with bevacizumab 2 and 4 weeks of treatment exposure	Lipophilic serum extracts: RCC treated and responder group, *n* = 10 RCC treated and non-responder group, *n* = 15	LC-MS T-distributed stochastic neighbor embedding Linear mixed effects models 10-Fold cross validation	Immunotherapy effects in RCC treated group: PC(38:0), PC(42:0), PC(42:2), PC(40:6), PC(42:3), PC(44:6), SM(OH, 22:1), SM(24:1), SM(26:1), SM(20:2)	Immunotherapy upregulated β- oxidation of lipids rather than glycolysis and altered T cell metabolism to enhance therapeutic response	[95]

786-O: hypertriploid renal cell carcinoma cell line; ^1^H NMR: proton nuclear magnetic resonance; CE: ceramide; CE-TOF MS: capillary electrophoresis-time of flight mass spectrometry; LC-MS: liquid chromatography-mass spectrometry; LC-MS/MS: liquid chromatography-tandem mass spectrometry; OPLS: orthogonal partial least squares; PCA: principal component analysis; PC: phosphocholine; RCC: renal cell carcinoma; SM: sphingomyelin.

## Data Availability

Not applicable.

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
