# Peer review of "Pharmacometabolomics Applied to Personalized Medicine in Urological Cancers"

_pharmaceuticals, 2022, doi:10.3390/ph15030295_

Round 1

Reviewer 1 Report

Dear Authors,

The article is very interesting and useful.

Major:

- Table 1: it is not clear the structure of the samples for the two cell lines (PC3 and LNCaP); abbreviations from the table should be explained below the table; 

- reference list is incomplete (missing references from 65 to 86)

Author Response

Response to reviewer 1 comments

Firstly, we would like to thank the Reviewer 1 for the comments and suggestions that helped to improve the quality of the manuscript. Below is an item-by-item reply to the Reviewer’s 1 comments and suggestions.

Comments and Suggestions for Authors

Dear Authors,

The article is very interesting and useful.

Major:

Table 1: it is not clear the structure of the samples for the two cell lines (PC3 and LNCaP); abbreviations from the table should be explained below the table.

Author’s reply

We thank the Reviewer for this comment. The structure of the samples of PC3 and LNCaP has been improved in Table 1 (page 9) and all abbreviations have been explained in table footnotes.

Reference list is incomplete (missing references from 65 to 86).

Author’s reply

References from 65 to 86 are listed in the page 26 of the manuscript.

Reviewer 2 Report

It’s a very informative review article on the application of pharmacometabolomic approaches to identify predictive biomarkers of response to hormone therapy, targeted therapy, immunotherapy, and chemotherapy for the most prevalent urological cancers The manuscript is well structured and written. References were properly cited as well.

However, I have few suggestions before publication:

Line 213: “pharmacometabolomics in biomarker discovery field relies on the assumption that drug exposure alters the metabolome”. The authors should consider include the potential effect of specific metabolic phenotypes on treatment response/resistance/toxicity.

Line 216: metabolite separation is not mandatory in all pharmacometabolomics studies

Line 261-272: It would be valuable to cite other statistical approaches for data analysis (e.g.: GSEA, hierarchical clustering, etc)

Line 277: Also, reference to pharamacometabolomic studies where different omics data are integrated should be discussed.

Tables: Although the review focuses on the application of pharmacometabolomic approaches to identify predictive biomarkers of response to hormoneherpy, targeted therapy, immunotherapy, and chemotherapy for the most prevalent urological caners” the information included in the Tables is focused, in most cases, on the metabolic alterations caused by the treatment, and limited information is included regarding predictive potential of these changes. This aspect should be further discussed/clarified.

Author Response

Response to reviewer 2 comments

Firstly, we would like to thank the Reviewer 2 for the comments and suggestions that helped to improve the quality of the manuscript. Below is an item-by-item reply to the Reviewer’s 2 comments and suggestions.

Comments and Suggestions for Authors

It’s a very informative review article on the application of pharmacometabolomic approaches to identify predictive biomarkers of response to hormone therapy, targeted therapy, immunotherapy, and chemotherapy for the most prevalent urological cancers The manuscript is well structured and written. References were properly cited as well.

However, I have few suggestions before publication:

Line 213: “pharmacometabolomics in biomarker discovery field relies on the assumption that drug exposure alters the metabolome”. The authors should consider include the potential effect of specific metabolic phenotypes on treatment response/resistance/toxicity.

Author’s reply

We thank the Reviewer for this suggestion, which has now been addressed in the manuscript as follows:

Page 5, line 213: “Since the primary goal of pharmacometabolomics in biomarker discovery field relies on the assumption that the metabolic fingerprint is altered by treatment response, resistance and/or toxicity, the workflow initiates with the collection of biological samples before (baseline) and after treatment.”

Line 216: metabolite separation is not mandatory in all pharmacometabolomics studies

Author’s reply

We thank the Reviewer for this comment that has now been considered in the manuscript as follows:

Page 5, line 216: “The second step in pharmacometabolomics pipeline is data acquisition that includes metabolite detection by high-throughput analytical techniques.”

Line 261-272: It would be valuable to cite other statistical approaches for data analysis (e.g.: GSEA, hierarchical clustering, etc)

Author’s reply

We appreciate the Reviewer's input, which has been incorporated into the manuscript as follows:

Page 6, line 261: “Hierarchical cluster analysis is also commonly used to build a tree diagram based on the similarity or differences between sample groups (clusters).”

Line 277: Also, reference to pharamacometabolomic studies where different omics data are integrated should be discussed.

Author’s reply

We understand the reviewer's point of view; however, the majority of pharmacometabolomic studies fail to combine multiple omics data sets. For the ones that included different omics approaches, a brief discussion has been inserted in the manuscript as follows:

Page 7, line 318-319: “The pharmacometabolomic analysis was complemented with Western blotting to confirm the inhibition of the target proteins.”

Page 8, line 353-355: “Moreover, a reduction on the expression of cell surface transporters necessary for glutamine intake was noted through Western blot assays.”

Page 15: “Expression of key proteins in tumor samples from patients receiving or not receiving the neoadjuvant therapy were also evaluated to corroborate the findings obtained by HPLC analysis.”

Page 21: “These results were corroborated by the higher expression of glutamine transporters in sunitinib-resistant cells.”

Page 22: “The prognostic value of the VLCFA-containing lipids was also investigated by transcriptomic analysis.”

Although the review focuses on the “application of pharmacometabolomic approaches to identify predictive biomarkers of response to hormone therapy, targeted therapy, immunotherapy, and chemotherapy for the most prevalent urological cancers” the information included in the Tables is focused, in most cases, on the metabolic alterations caused by the treatment, and limited information is included regarding predictive potential of these changes. This aspect should be further discussed/clarified.

Author’s reply

We thank the Reviewer for this comment that has been considered now in the manuscript. The articles published so far on this topic addressed the effects of treatments on the body (changes in metabolite levels), but do not validate their applicability as predictive biomarkers. Nonetheless, the reviewer's concern has been addressed in the revised manuscript as follows:

Page 1, Line 24: “Hence, this review focuses on the application of pharmacometabolomic approaches to identify the metabolic responses to hormone therapy, targeted therapy, immunotherapy, and chemotherapy for the most prevalent urological cancers.”

Page 6, Line 288: “In this section, a review of the literature was performed to access studies that apply pharmacometabolomic approaches to identify the metabolic responses to hormone therapy, immunotherapy, targeted therapy, and chemotherapy in PCa, BCa, and RCC.”

Page 22: “Further studies are needed to validate the impact of the different treatments on the metabolic fingerprint in larger and independent clinical cohorts of different populations (e.g., ethnicities) of urological cancers patients, as well as to complement those studies with multi-omics analyses to improve accuracy in predictive biomarker detection and a better understanding of the mechanisms underlying cancer therapeutic resistance.”

Round 2

Reviewer 1 Report

Dear authors,

The manuscript is clear now from my point of view and I don't have any comments now.  

Reviewer 2 Report

I think all concerns have been addressed